# Efficacy of Different Bacillus of Calmette-Guérin (BCG) Strains on Recurrence Rates among Intermediate/High-Risk Non-Muscle Invasive Bladder Cancers (NMIBCs): Single-Arm Study Systematic Review, Cumulative and Network Meta-Analysis

**DOI:** 10.3390/cancers15071937

**Published:** 2023-03-23

**Authors:** Francesco Del Giudice, Vincenzo Asero, Eugenio Bologna, Carlo Maria Scornajenghi, Dalila Carino, Virginia Dolci, Pietro Viscuso, Stefano Salciccia, Alessandro Sciarra, David D’Andrea, Benjamin Pradere, Marco Moschini, Andrea Mari, Simone Albisinni, Wojciech Krajewski, Tomasz Szydełko, Bartosz Małkiewicz, Łukasz Nowak, Ekaterina Laukhtina, Andrea Gallioli, Laura S. Mertens, Gautier Marcq, Alessia Cimadamore, Luca Afferi, Francesco Soria, Keiichiro Mori, Karl Heinrich Tully, Renate Pichler, Matteo Ferro, Octavian Sabin Tataru, Riccardo Autorino, Simone Crivellaro, Felice Crocetto, Gian Maria Busetto, Satvir Basran, Michael L. Eisenberg, Benjamin Inbeh Chung, Ettore De Berardinis

**Affiliations:** 1Department of Maternal-Infant and Urological Sciences, Policlinico Umberto I Hospital, “Sapienza” University of Rome, 00185 Rome, Italy; 2Department of Urology, Stanford University School of Medicine, Stanford, CA 94305-5101, USA; 3Department of Urology, Comprehensive Cancer Center, Medical University of Vienna, 1030 Vienna, Austria; 4Department of Urology, La Croix du Sud Hospital, 31130 Quint-Fonsegrives, France; 5Department of Urology and Division of Experimental Oncology, Urological Research Institute, Vita-Salute San Raffaele, 20132 Milan, Italy; 6Department of Experimental and Clinical Medicine, University of Florence—Unit of Oncologic Minimally-Invasive Urology and Andrology, Careggi Hospital, 50134 Florence, Italy; 7Urology Unit, Department of Surgical Sciences, Tor Vergata University Hospital, University of Rome Tor Vergata, 00133 Rome, Italy; 8Department of Minimally Invasive and Robotic Urology, Wrocław Medical University, 50-367 Wrocław, Poland; 9Institute for Urology and Reproductive Health, Sechenov University, 119435 Moscow, Russia; 10Department of Urology, Fundacio Puigvert, 16444 Barcelona, Spain; 11Department of Urology, The Netherlands Cancer Institute, 1066 Amsterdam, The Netherlands; 12Urology Department, Claude Huriez Hospital, CHU Lille, 59000 Lille, France; 13Cancer Heterogeneity Plasticity and Resistance to Therapies, UMR9020-U1277—CANTHER, Institut Pasteur de LilleCHU Lille, Inserm, CNRS University of Lille, 59000 Lille, France; 14Department of Medical Area (DAME), Institute of Pathological Anatomy, University of Udine, 33100 Udine, Italy; 15Department of Urology, Luzerner Kantonsspital, 6004 Luzern, Switzerland; 16Urology Division, Department of Surgical Sciences, University of Studies of Torino, 10124 Turin, Italy; 17Department of Urology, The Jikei University School of Medicine, Nishi-Shimbashi, Minato-ku, Tokyo 105-8461, Japan; 18Department of Urology and Neurourology, Marien Hospital Herne, Ruhr-University Bochum, 44780 Herne, Germany; 19Department of Urology, Comprehensive Cancer Center Innsbruck, Medical University of Innsbruck, 6020 Innsbruck, Austria; 20Division of Urology, European Institute of Oncology, IRCCS, 20141 Milan, Italy; 21Department of Simulation Applied in Medicine, The Institution Organizing University Doctoral Studies (I.O.S.U.D.), George Emil Palade University of Medicine, Pharmacy, Sciences, and Technology, 540142 Târgu Mureș, Romania; 22Department of Urology, Rush University Medical Center, Chicago, IL 60612, USA; 23Health Sciences System, Department of Urology, University of Illinois Hospital e Camp, Chicago, IL 60612, USA; 24Reproductive Sciences and Odontostomatology, Urology Unit, Department of Neurosciences, University of Naples “Federico II”, 80138 Naples, Italy; 25Department of Urology and Organ Transplantation, University of Foggia, 71122 Foggia, Italy

**Keywords:** bladder cancer, non-muscle invasive bladder cancer, recurrence rate, BCG immunotherapy, BCG strain, network meta-analysis

## Abstract

**Simple Summary:**

Bacillus of Calmette-Guérin (BCG) is the gold standard as per adjuvant intravesical treatment for intermediate and high-risk non-muscle invasive bladder cancer (NMIBC). Nevertheless, drug-related toxicity, compliance, and a shortage of BCG availability make the completion of the planned treatment schedule challenging in many patients, thus possibly impacting survival outcomes. No one specific BCG strain out of the several available ones worldwide has so far demonstrated its superiority profile in prolonging time to recurrence and progression. In our systematic review and network meta-analysis, we compared to most widely adopted BCG strains and demonstrated that BCG strain Tice, RIVM, and Tokyo 172 could display potential enhanced benefits, thus possibly supporting the use of such strains for future BCG trials in NMIBCs.

**Abstract:**

Background: In an era of Bacillus of Calmette-Guérin (BCG) shortages, the comparative efficacy from different adjuvant intravesical BCG strains in non-muscle invasive bladder cancer (NMIBC) has not been clearly elucidated. We aim to compare, through a systematic review and meta-analysis, the cumulative BC recurrence rates and the best efficacy profile of worldwide available BCG strains over the last forty years. Methods: PubMed, Scopus, Web of Science, Embase, and Cochrane databases were searched from 1982 up to 2022. A meta-analysis of pooled BC recurrence rates was stratified for studies with ≤3-y vs. >3-y recurrence-free survival (RFS) endpoints and the strain of BCG. Sensitivity analysis, sub-group analysis, and meta-regression were implemented to investigate the contribution of moderators to heterogeneity. A random-effect network meta-analysis was performed to compare BCG strains on a multi-treatment level. Results: In total, n = 62 series with n = 15,412 patients in n = 100 study arms and n = 10 different BCG strains were reviewed. BCG Tokyo 172 exhibited the lowest pooled BC recurrence rate among studies with ≤3-y RFS (0.22 (95%CI 0.16–0.28). No clinically relevant difference was noted among strains at >3-y RFS outcomes. Sub-group and meta-regression analyses highlighted the influence of NMIBC risk-group classification and previous intravesical treated categories. Out of the n = 11 studies with n = 7 BCG strains included in the network, BCG RIVM, Tice, and Tokyo 172 presented with the best-predicted probability for efficacy, yet no single strain was significantly superior to another in preventing BC recurrence risk. Conclusion: We did not identify a BCG stain providing a clinically significant lower BC recurrence rate. While these findings might discourage investment in future head-to-head randomized comparison, we were, however, able to highlight some potential enhanced benefits from the genetically different BCG RIVM, Tice, and Tokyo 172. This evidence would support the use of such strains for future BCG trials in NMIBCs.

## 1. Introduction

The standard of care for intermediate and high-risk non-muscle invasive bladder cancer (NMIBC) status post trans-urethral resection of bladder tumor (TURBT) with or without secondary resection (Re-TUR) is represented by an adjuvant induction course of intravesical Bacillus of Calmette-Guérin (BCG) followed by an adequate maintenance schedule for at least one year [1]. Additionally, Randomized Controlled Trial (RCT) evidence has suggested that the three years of the maintenance schedule originally described by Lamm et al. (i.e., a total of n = 27 intravesical instillations within 3 years) was able to provide an additional reduction of recurrence in high-risk cases, but there was no difference in progression and cancer survival when the data was examined over the initial one-year period [2]. Of note, this was not similarly true for intermediate risk tumors [3]. Nevertheless, drug-related toxicity, compliance, and a shortage of BCG availability make the completion of the planned treatment schedule challenging for many patients. In particular, BCG production and global distribution remain particularly fraught. The currently available strains adopted for NMIBC are indeed the result of years of increasing genotypic diversity from the original mycobacterium bovis seed released in 1921 by Calmette and Guérin as per the tuberculosis vaccine [4,5]. The subsequent end of production of the BCG Connaught strain in 2012 resulted in a global shortage of BCG, for which European Association of Urology (EAU) Guidelines developed a statement by the NMIBC Panel exploring the possible available options including reduced maintenance schedule and frequency or alternative treatments such as device-assisted chemotherapy instillations (microwave-induced hyperthermia [RITE] or electromotive drug administration [EMDA]) [6]. Unfortunately, in most cases, these options are experimental or less effective while significantly more expensive, highlighting the issues of BCG shortages [7,8].

According to the literature, there is evidence that genetically distinct BCG strains have been associated with differences in immune responses, providing possible differences in recurrence rate outcomes [4]. Despite this evidence, the currently available EAU Guidelines do not offer any conclusive indications as to individual BCG strains and their impact on anti-tumor efficacy.

One relatively recent meta-analysis that compared different BCG strains was published in 2017 by Boehm et al. [9]. Although the tremendous effort endured by the authors, the outcomes presented were mainly made with intravesical chemotherapy as the common comparator for the expression of the effect size estimators as well on BCG naïve population and with no clear distinction on the different available time-dependent recurrence rate endpoints commonly adopted by the NMIBC series. Additionally, in the last 5 years, there has been renewed interest in retrospective cohort studies which assess this unresolved topic, making an updated analysis on the topic especially germane.

Therefore, our aim was to provide an up-to-date review of the literature on the topic of BCG strain comparison by conducting a cumulative meta-analysis of event rates for bladder cancer recurrence assessed at different recurrence-free survival (RFS) endpoints. Accordingly, we developed a systematic review and meta-analysis comparing the effectiveness of different treatments of the most commonly adopted BCG strains, with varying doses and schedules accounting for the most known and validated NMIBC survival confounders. We also conducted a network meta-analysis, including those studies that adopted different BCG strains on a multi-treatment level, to explore the compared efficacy of one specific strain over the other. Furthermore, we aimed to be as inclusive as possible in our inclusion criteria to assess and balance the relative effect of different BCG schedules, dosages, and a variety of clinical scenarios.

## 2. Methods

This systematic review and network meta-analysis was conducted according to Preferred Reporting Items for Systematic Reviews and Meta-Analyses (PRISMA) guidelines [10]. A research question was established based on the Patient-Index test-Comparator-Outcome-Study design (PICOS) criteria as the following: is there any superiority in terms of efficacy profile for preventing bladder cancer recurrence rate amongst different available BCG strains?

Furthermore, our goal was to compare current evidence within all the available retrospective/prospective and/or single-/multicenter cohort studies applying different BCG strains in the adjuvant setting of patients treated by therapeutic and/or sampling trans-urethral resection (TUR) followed or not followed by secondary resection (Re-TUR). In particular, we determined the pooled cumulative recurrence rate and a network multi-treatment comparison meta-analysis among all the available BCG strains utilized worldwide over the last forty years. Finally, our review has been revised and approved by PROSPERO with the following reference number: CRD42022380372.

### 2.1. Evidence Acquisition

We performed a systematic review of the literature in PubMed, Scopus, Web of Science, Embase, and Cochrane from 1982 to November 2022 to identify studies that examined the implementation of intravesical BCG immunotherapy for the adjuvant treatment of NMIBC and evaluated the pooled event rate trends and network indirect/direct comparison between the available series. The reference lists of the included studies were also screened for relevant articles. Both original prospective and retrospective cohort studies were included and critically evaluated (Level of Evidence: II and III-a), as well as randomized controlled trials (RCTs) focused on different BCG outcomes with comparisons on schedule, dose, and concomitant additional therapies where the type of strain administered was specified. Case reports, abstracts, and meeting reports were excluded from the analysis. Search terms included, but were not limited to: bladder cancer, AND non-muscle invasive bladder cancer AND adjuvant BCG immunotherapy or intravesical BCG implementation AND BCG strain or sub-strain AND efficacy profile AND recurrence rate AND; secondary fields: non-muscle invasive bladder cancer and recurrence rate; BCG strain and efficacy profile; BCG intravesical immunotherapy and schedule; BCG intravesical immunotherapy and dose. A comprehensive list of primary and secondary fields of search criteria has been presented in Appendix A.

### 2.2. Selection of the Studies and Criteria of Inclusion

Entry into the analysis was restricted to data from original articles that examined patients with primary and/or recurrent NMIBC diagnoses, classified as intermediate- and/or high-risk according to the European Organization for Research and Treatment of Cancer (EORTC) and/or EAU groups and that reported the rate of BC recurrences over the course of variable follow-up time after the administration of intravesical BCG immunotherapy. In addition, only those studies which included data to reconstruct the number of the recurrence events assigned per each different BCG strain and that was structured on a single or multiple comparison arm level were considered suitable for further consideration. In the multi-comparison experiences, studies were considered eligible if at least one of the involved study arms declared the strain of BCG adopted regardless of the schedule, dosage, and BC history of the participants.

Articles were excluded if they met one or more of the following criteria: inadequate information for data extraction or quality assessment; inclusion of study population consisting of <10 patients; presented outcomes which dealt with other topics (e.g., BCG non for BC immunotherapy, BCG, and other outcomes rather than recurrence such as BC progression and/or side effects).

Five authors (FDG, VA, EB, CMS, EDB) independently screened the titles and abstracts of all articles using predefined inclusion criteria. The full-text articles were examined independently by the five (FDG, VA, EB, CMS, EDB) to determine whether or not they met inclusion criteria. Final inclusion was determined by the consensus of all investigators. Selected articles were then critically analyzed. The following data were extracted from the included studies by using a standardized form: the origin of study (institution and period of enrollment), size of the study population, period of time prospectively/retrospectively covered (i.e., Recurrence-free survival [RFS] endpoints, mean/median follow-up time), previous BC history (only primary, only recurrent, mixed primary/recurrent NMIBCs), previous intravesical therapy (i.e., previous intravesical chemotherapy [CHT] and/or BCG naïve or already BCG treated), EAU risk group (only intermediate, only high-risk, mixed intermediate/high-risk), schedule of BCG (only induction, induction plus maintenance), BCG dose (i.e., full, half, one-third). Finally, baseline clinical and demographic patient information and tumor features (e.g., mean/median age, range of patients, gender representation, percentage of smoking, etc.) were annotated.

### 2.3. Assessment of Quality for Studies Included and Statistical Analysis

To assess the risk of bias (RoB), all included experiences were independently reviewed using the “*Quality Assessment Tool for Observational Cohort and Cross-Sectional Studies*” provided by the National Health Institute (NIH) [11]. Biases screened included selection bias, information bias, and measurement bias or confounding bias (including cointerventions, differences at baseline in patient characteristics, etc.). Studies were rated as poor quality, fair or good, with higher RoB leading to poor quality (“−”) ratings and low RoB leading to good quality (“+”) ratings. Publication bias was tested both by visual assessment of the Deeks’ funnel plot and calculation of the p-value using the Deeks’ asymmetry test [12]. The ‘*Trim and Fill*’ method was implemented to explore the possible nature of studies “missed” in the review [13]. Statistical analyses, along with reporting and interpretation of the results, were conducted according to the previously described methodology [14,15,16,17] and consisted of two analytical steps.

First, a conventional meta-analysis of the pooled event rate (i.e., BC recurrence rate) and 95% confidence intervals (CIs) was performed using random effect according to *DerSimonian–Laird* method [18]. Sensitivity analyses were performed to assess the contribution of each study to the pooled estimate by excluding individual trials one at a time and recalculating the pooled estimates for the remaining studies (leave-one-out meta-analysis). Evaluation for the presence of heterogeneity was done using [12]: (I) Cochran’s Q-test with *p* < 0.05 signifying heterogeneity; (II) Higgins I^2^ test with inconsistency index (I^2^) = 0–40%, heterogeneity might not be important; 30–60%, moderate heterogeneity; 50–90%, substantial heterogeneity; and 75–100%, considerable heterogeneity. Our results are graphically displayed as forest plots on a per-single study arm level, with pooled results indicating the overall BC recurrence rate across each series implementing different types of BCG strains. Subgroup analyses were performed looking at differences in categorical confounders (e.g., EAU risk group, BC history, study design, BCG schedule, BCG dosage, prev. intravesical CHT, etc.). Meta-regression analyses were performed using available continuous variables retrieved from the studies. Pooled weighted estimates were plotted against the following available quantitative variables: mean/median age of the patients, the total number of patients study arm patients, range of study time screened (months retrospectively or prospectively imputed), the relative percentage gender distribution as well as smoking and BCG-specific study characteristics.

Second, a random-effect network meta-analysis within a Bayesian framework [19] was performed to evaluate the relative efficacy of the available studies comparing different intravesical BCG strains on a multi-treatment level. Odds ratios (ORs) with 95% CI of each contrast between strains on the outcome of BC recurrence were displayed. The surface under the cumulative ranking curve (SUCRA) [20] was used to estimate the ranking probabilities for the different BCG stains in order to obtain a treatment hierarchy for the outcome of BC recurrence. An absolute measure of fit residual deviance was considered to formally check the model’s overall fit [21]. The presence of inconsistency in network meta-analysis was evaluated by a loop-specific approach [22]. Inconsistency with 95% CIs between direct and indirect analysis for the comparison of each outcome was calculated to assess the presence of inconsistency in each loop. Inconsistency was defined as a disagreement between direct and indirect evidence with a 95% CI excluding 0.

Calculations were accomplished using the ‘*meta*’ and ‘*network setup*’ packages on Stata version 17.1 (Stata Corporation, College Station, TX, USA).

## 3. Results

### 3.1. Search Results

The initial search yielded n = 282 articles (PubMed: 258; Cochrane: 17; and Embase: 7). n = 57 were excluded as they contained overlapping data or were duplicates appearing in multiple databases. Of the remaining n = 220, n = 25 were further excluded because they did not examine BCG therapy as per adjuvant intravesical use in NMIBCs (n = 23), did not report information regarding the type of BCG strain adopted (n = 27) or were review papers, editorials, or abstracts (n = 43). Full-text articles were then reevaluated and critically analyzed for the remaining n = 125 journal references. Within this in-depth review, a further n = 63 did not meet the inclusion criteria. The remaining n = 62 studies were included in the quantitative analysis (Appendix A). No study was considered to be seriously flawed, and performance bias was overall low, with some attrition bias due to incomplete outcome data across all the studies. Individual RoB, as well as visual assessment of the Deeks’ funnel plots, are illustrated in Appendix A, respectively.

### 3.2. Characteristics of The Populations, Study Design, and Location

Sixty-two studies [2,3,23,24,25,26,27,28,29,30,31,32,33,34,35,36,37,38,39,40,41,42,43,44,45,46,47,48,49,50,51,52,53,54,55,56,57,58,59,60,61,62,63,64,65,66,67,68,69,70,71,72,73,74,75,76,77,78,79,80,81,82] were included in the systematic review with a total of n = 15,412 patients who received adjuvant intravesical BCG following therapeutic and/or sampling TUR ± Re-TUR with EAU intermediate/high-risk NMIBCs. The number of single study arms was n = 100 [2,3,23,24,25,26,27,28,29,30,31,32,33,34,35,36,37,38,39,40,41,42,43,44,45,46,47,48,49,50,51,52,53,54,55,56,57,58,59,60,61,62,63,64,65,66,67,68,69,70,71,72,73,74,75,76,77,78,79,80,81,82], including a total of n = 10 [2,3,23,24,25,26,27,28,29,30,31,32,33,34,35,36,37,38,39,40,41,42,43,44,45,46,47,48,49,50,51,52,53,54,55,56,57,58,59,60,61,62,63,64,65,66,67,68,69,70,71,72,73,74,75,76,77,78,79,80,81,82] different BCG strains adopted. A comprehensive list of study characteristics is presented in Table 1. The study period was from 1982 to 2022, including both male and female subjects with mean age comprised of 53 to 77 who had been followed for a median time of 43 months (range, 12–108 months). The vast majority of the studies (n = 34, [24,25,26,27,28,29,34,35,36,37,38,39,40,41,42,48,50,51,53,55,56,60,62,63,65,68,69,71,75,76,77,79,80,81]) were single-arm with no direct comparison across other strains. On the contrary, the remaining n = 27 [2,3,23,30,31,32,33,43,44,45,47,49,52,54,57,58,59,61,64,66,67,70,72,73,74,78,82] were dual or multi-treatment comparison analysis. The cumulative sample size for each study arm was directly associated with the study design with n = 79 [2,3,23,24,25,26,27,28,29,34,35,36,37,38,39,40,41,42,43,44,45,46,47,48,49,50,51,52,53,54,55,56,57,58,60,62,64,65,67,68,69,70,71,72,73,74,75,76,79,80,82] prospective cohort analysis including from a minimum to a maximum of n = 9 and n = 816 patients respectively. Additionally, out of the prospective series, n = 12 were head-to-head randomized controlled comparisons yet not primarily focused on strain comparison but rather assessing differences in BCG maintenance schedule (n = 7, [2,44,45,47,52,64,82]), dose (n = 4, [3,23,54,74]) or additional related adjuvant therapies (n = 1), [72]). Retrospective series were in total n = 21 [30,31,32,33,59,61,63,66,77,78,81], accounting for a consistently larger median sample (from n = 27 to n = 1142) and investigating all the multi-treatments BCG comparisons (n = 5, [30,32,33,59,78]). Finally, the systematic review included worldwide experiences, with the vast majority of the studies deriving from Europe (n = 20), followed by Japan (n = 12) and the USA (n = 8), while in several cases (n = 12) there a multicenter design was implemented with no primary institution contribution detectable (Table 1).

### 3.3. BCG-Specific Characteristics: Stains, Schedule, and Dose

The type of BCG schedule administration retrieved accounted for study arms assessing induction only (n = 39 [2,26,31,34,38,40,42,43,44,45,46,47,51,52,55,58,61,64,67,68,69,70,71,72,75,76,77,80,82]) and induction followed by any type of maintenance schedule (n = 61 [2,3,23,24,25,27,28,29,30,31,32,33,35,36,37,39,41,44,45,47,48,49,50,52,53,54,56,57,59,60,61,62,63,64,65,66,70,71,72,73,73,74,74,75,76,77,78,81,82]).

Similarly, the majority of the single study arm based their comparison on a full BCG dose (n = 87 [2,3,23,24,25,26,27,28,29,30,32,34,35,36,37,38,40,41,42,43,44,45,46,47,48,49,50,51,52,53,54,56,57,58,59,61,62,63,64,65,66,67,68,69,70,71,72,73,74,75,76,77,79,80,81,82]) followed by half-dose n = 2 [31,49] with only n = 9 [3,48,53,54,60,74] experiences that fractioned the BCG administration to one-third of the original dose.

BCG Connaught was the most represented strain adopted, with a total sample of the patient included in this group accounting for n = 6960 patients. Overall, n = 23 [2,26,29,32,34,35,36,39,42,43,44,45,49,54,60,63,64,67,70,74,78,80,81] studies and n = 33 single-arm comparisons were based on its utilization. Moreover, before the suspension of its production happened in 2012, this strain was also the most used from the 2000s onwards. This strain was also significantly implemented in prospective longitudinal cohort studies with only n = 4 [33,64,79,82] retrospective series maintaining a good balance regarding the type of schedule and dose administered.

BCG Tice was the second most represented strain adopted, accounting for a total of n = 5326 patients. Overall, n = 21 [3,25,27,28,30,32,33,41,53,56,57,59,62,65,66,67,74,75,78,79,82] studies and n = 27 single-arm comparisons were based on its utilization. This strain was mostly studied in prospective longitudinal cohort studies with only n = 6 [30,32,33,59,66,78] retrospective series with a predominance of induction plus maintenance schedule and full dose administered.

For the BCG Tokyo 172 strain, there were a total of n = 726 patients, with all the studies deriving from Japan. Overall, n = 11 [31,46,49,52,61,69,70,71,72,76,77] studies and n = 16 single-arm comparisons were analyzed. This strain was also significantly utilized in prospective longitudinal cohort studies with only n = 5 [31,61,77] retrospective series maintaining a good balance regarding the type of schedule, with a full dose mostly administered.

The Pasteur strain accounted for a total of n = 8 [24,47,48,50,55,57,58,68] studies. All the studies included in this subgroup were prospective, considering a full-dose BCG treatment, with a good balance regarding the type of schedule and only full dose adopted.

Regarding the remaining series and related BCG stains, there were n = 11 [23,30,32,33,37,38,48,51,58,59,73] studies and n = 15 [23,30,32,33,37,38,48,51,58,59,73] single-arm comparisons for a total of further n = 2021 patients. Six strains were mentioned in the studies, specifically RIVM (n = 5 [32,33,37,38,59]), Danish 1331 (n = 3 [23,51]), Moreau (n = 2 [30,59]), Glaxo (n = 2 [58]), Montreal Armand Frappier (n = 1 [48]) and Sii Onco BCG (n = 2 [73]). These strains were mostly studied in prospective longitudinal cohort studies with only n = 4 [30,32,33,59] retrospective series.

#### 3.3.1. BC Recurrence Rate Meta-Analysis

From the n = 62 studies analyzed [2,3,23,24,25,26,27,28,29,30,31,32,33,34,35,36,37,38,39,40,41,42,43,44,45,46,47,48,49,50,51,52,53,54,55,56,57,58,59,60,61,62,63,64,65,66,67,68,69,70,71,72,73,74,75,76,77,78,79,80,81,82], the cumulative BC recurrence rate within the n = 100 single-arm assessed varied from 0.08 (95%CI −0.06–0.24) to 0.88 (95%CI 0.57–1.19) irrespective of any confounders, sub-group, or BCG-specific covariates. As expected, at this preliminary assessment, there was substantial heterogeneity across every single study arm with I^2^ 82.67%, Q (101): 583.26, *p* < 0.001. Publication bias was initially assessed by Galbright and Funnel plot (Appendix A). The inspection of both plots suggested that there was no small-study effect, with the smaller studies tending to have higher recurrence rate estimates, suggesting the absence of publication bias (Egger test, *p* = 0.19). Additionally, the “Trim and Fill” method suggested that only n = 2 studies would have needed to be included to remove residual asymmetry from the funnel plot. The main contribution to single-arm variability was identified by both demographic cohort imbalances and clinical BC characteristics and, most importantly, by BCG-specific confounders such as strains, administration schedule, dosage, as well as RFS time points, and study design. A detailed list of sub-group analyses is presented in (Appendix A). The total number of BCG strains assessed was n = 10. Nevertheless, given the observed wide variability in the influence of those BCG strains studies with no more than n = 150 patients per study arm on pooled estimates, BCG Danish 1331 (pts. n = 130), Glaxo (pts. n = 24), Montreal Armand Frappier (pts. n = 17), and Sii-onco (pts. n = 87) were excluded from the cumulative recurrence rate meta-analysis. Additionally, since the well knows the influence of the time-dependent effect on BC recurrences by predefined RFS cut-off within each study, the results had been reported stratified by ≤3-y and >3-y RFS timepoints, respectively. Such criterion is derived, on the one hand, from the common literature-based RFS endpoints for NMIBCs usually set at 3-y and 5-y and, on the other hand, from the observation of non-clinically relevant variability observed within these predefined ranks across the studies enclosed.

#### 3.3.2. BC Recurrence Rate by ≤3-y RFS Endpoints and BCG Strain

N = 2 BCG strains were evaluated by only n = 2 study arms [23,73], as well as the study of Di Lorenzo et al. [35] was the sole significantly affecting the heterogeneity statistic, and these were therefore removed from the analysis. Thus, the sub-sample of 1 up to 3 y RFS included n = 33 studies [31,39,40,42,43,44,45,46,47,49,52,53,55,57,58,63,65,66,68,70,71,74,79,80,82] assessing n = 4 BCG strains (i.e., Connaught, Pasteur, Tice, and Tokyo 172) and n = 5123 patients accounting for a cumulative recurrence rate of 0.31 (95%CI 0.28–0.35). This data clustering led to a significant reduction in the overall in-studies heterogeneity, now accounting for I^2^ 68.94%, Q (95): 138.46, *p* = 0.001. At this time, an inspection of the funnel plot suggested that there was no small-study effect, with the smaller studies tending to have higher recurrence rate estimates, suggesting an absence of publication bias (Egger test, *p* = 0.67). The “Trim and Fill” method suggested that no “missing” studies needed to be included to remove any asymmetry from the funnel plot (Appendix A). Residual inconsistency was found to be attributable to previous BC history, previous BCG treatment, and BCG strains adopted, as detailed in the list of sub-group analysis Figure 1a. On meta-regression analysis, we found no significant nor clinical correlation among all quantitative moderators assessed in the study except from the increasing relative percentages of smokers across the studies reporting the information (Coeff: 0.01, SE: 0.004, *p* = 0.006) and the percentages of previously BCG treated patients (Coeff: 0.002, SE: 0.0009, *p* = 0.004) as depicted in Appendix A.

A close focus on BCG strains across the ≤3-y RFS studies is shown in Figure 1b as an incremental rate by BC recurrence rate and in Figure 1c as a cumulative meta-analysis by publication year. BCG Tokyo 172 showed the lowest recurrence rate with a pooled estimate of 0.22 (95%CI 0.16–0.28) and a reduction of the observed cumulative percentage of recurrence events over the years. The pooled rate of BC recurrence was slightly higher and overlapping for both BCG Connaught and Pasteur (i.e., 0.30, 95%CI 0.26–0.34 and 0.28, 95%CI 0.19–0.38, respectively) while pooled rate observed for BCG Tice was 0.39, 95%CI 0.32–0.45). Interestingly, except from BCG Tokyo 172, we registered a slight but consistent increase in the recurrence rate by cumulative meta-analysis sorted by publication year, possibly indicating a trend toward reduction in efficacy over the years of the administration of such strains.

#### 3.3.3. BC Recurrence Rate for >3-y RFS Endpoint and BCG Strain

Due to the significantly reduced sample size, the studies of Mukherjee et [58] and Jarvien et al. [50] were significantly affecting the heterogeneity statistic and were removed from the analysis. Moreover, BCG Moreau was only represented by the study arm of Nowak et al. [59] and thus was excluded from the analysis. Finally, one of the study arms from Herr et al. [44], which accounted for only recurrent BC patients, was removed as per sensitivity analysis. Thus, the updated sub-group of studies reporting more than 3-y RFS timepoints accounted for n = 26 studies assessing in total n = 4 BCG strains (i.e., Connaught, RIVM, Tice, and Tokyo 172) with a cumulative sample size of n = 9207 patients distributed in n = 40 study arms and showing a pooled recurrence rate of 0.40 (95%CI 0.36–0.43). The overall measured study arm heterogeneity resulted, however substantial, with I^2^ 85.93%, Q (95): 277.1, *p* < 0.001. The rest of the within-study variability for recurrence rate was mainly due to already intravesical CHT-treated patients (0.36, 95%CI 0.31–0.41 vs. 0.48, 95%CI 0.44–0.58) with no further influence from the previously mentioned sub-group category, not even BCG strains. Detailed sub-groups analysis is presented in Figure 2a. Inspection of the funnel plot revealed no small-study effect, with the smaller studies tending to have higher recurrence rate estimates, suggesting an absence of publication bias (Egger test, *p* = 0.99). Nevertheless, the “Trim and Fill” method suggested that n = 6 “missing” studies would have needed to remove the visual asymmetry from the funnel plot, thus justifying part of the observed heterogeneity (Appendix A). Additionally, the residual heterogeneity was further explored by meta-regression analysis (Appendix A) and once again identified primarily the total sample groups distribution and the relative percentage of previous intravesical CHT administration with direct incremental correlation with the effects sizes (Coeff: 0.0001, SE: 0.000052, *p* = 0.001, and Coeff: 0.0048, SE: 0.0022, *p* = 0.032 respectively). Interestingly, male and female gender revealed an inversed relationship with the effect size (Coeff: −0.0075, SE: 0.0024, *p* = 0.002, and Coeff: 0.004, SE: 0.0014, *p* = 0.006), which is in line with sex-driven differences in NMIBC recurrence rate outcomes over long term follow-up. Finally, as expected, increasing percentages of subjects who had previously undergone BCG were significantly associated with an increased rate of recurrences over time (Coeff: 0.0032, SE: 0.0009, *p* = 0.001).

At long-term RFS endpoints, there were no clinical nor significant differences in the effect sizes for the n = 4 BCG strains analyzed (Figure 2b). Similar to the previous analysis, BCG Tice showed a slight yet non-significant higher reported recurrence rate when compared with BCG Connaught, RIVM, and Tokyo 172. This last strain also presented here the lowest pooled recurrence rate (0.35, 95%CI 0.21–0.48) yet the highest inter-strain study heterogeneity I^2^ 78.61%, Q (95): 28.05, *p* < 0.001. Finally, each BCG strain assessed throughout cumulative meta-analysis by year of publication demonstrated a constant and significant trend towards a reduction in efficacy, confirming the same trajectory observed at shorter RFS endpoints (Figure 2c).

#### 3.3.4. Evidence Structure of Network Meta-Analysis for BCG Strain Comparison

In total, n = 11 [30,32,33,49,57,58,59,67,70,74,78] studies accounting for n = 7 different BCG strains assessed across n = 24 study arms met the inclusion criteria and were reviewed for the network mixed-treatment meta-analysis of BC recurrence. The range of study time was wide, accounting for a total of n = 5549 patients treated with intravesical BCG from 1992 to 2022. Of note, the study from Del Giudice et al. [32] and Nowak et al. [59] were the sole experiences directly comparing three strains at once, while the rest was focused on dual comparison within two strains. The network map on multiple comparisons is shown in Appendix A. Most of the studies focused on the comparison with BCG Tice. This strain had the highest number of interactions being adopted by n = 8 studies within a range of 26 years, followed by BCG Connaught (n = 6 interactions), BCG RIVM (n = 3 interactions), and BCG Moreau, Pasteur, Tokyo 172 (n = 2 interactions) with BCG Glaxo being adopted only in n = 1 comparison. Half of the studies were conducted in a retrospective design with a consistently larger total sample size reviewed, varying from a minimum of n = 112 patients to a maximum of n = 1142 in the treatment arms of Del Giudice et al. [32] and Witjes et al. [78] respectively. As expected, this was inversely represented when the design was prospective with per single study arm ranging between n = 9 and n = 325 in the study of Mukherjee [58] and Steinberg [74], respectively. The vast majority of the arms from the studies included in the network meta-analysis were homogeneous in the assessment of patients presenting with mixed EAU intermediated/high-risk NMIBCs (n = 17 [30,32,49,57,58,70,74,78]), that received a course of induction followed by maintenance schedule (n = 18 [30,32,33,49,57,59,74,78]), and with the administration of a full BCG dose (n = 23 [30,32,33,49,57,58,59,67,70,74,78]) throughout predefined timepoints for RFS set at 5-y (n = 16 [32,33,34,58,59,67,70,78]). Finally, there was more heterogeneity in the distribution of intravesically naïve arms both for previous BCG and intravesical CHT, respectively (n = 13 [32,33,49,59,67,70,74]).

#### 3.3.5. Network Meta-Analysis for Risk of BC Recurrence among Intravesical BCG Strains

Given its wider adoption over the range of study time, BCG Tice was considered for predefined reference (Appendix A). All the mixed comparisons were non-significant in the efficacy profile for the risk of BC recurrence. However, all the strains provided a trend towards significance in line with reported variability across literature assessing the BCG strain’s efficacy. These findings were therefore implemented to explore the probabilities that each treatment was the best under the consistency model as depicted in the rankogram of the different BCG stains in Appendix A. RIVM had a 27.2% cumulative probability of being the best BCG strain as well as the 30.6% and 21.8% of being the second and third best, respectively. Overall, this was followed by BCG Tice (12.5%, 31.7%, and 32.4%) and BCG Tokyo 172 (21.9%, 15.7%, and 14.7%). These assumptions were further confirmed by testing for the inconsistency of the displayed model (*p* = 0.11). The individual study results, grouped by treatment contrast and design, are displayed in Figure 3. All the assessed comparisons crossed the reference line yet with some interesting insight. BCG Connaught resulted superior to Tice, but this was true only in those treatment contrast where the BCG induction was not followed by any maintenance schedule [67]. The opposite was indeed found when an adequate BCG schedule was implemented [32,74,78]. Finally, in the investigation of the contrast between RIVM vs. Tice and Moreau vs. RIVM and Tice, there were overlapping results in the first case and a tendency favoring Moreau over the last two strains. These results were, in conclusion, validated by further exploring inconsistency by side-splitting of each node of the contrast comparison. The findings once again supported the consistent results aforementioned.

## 4. Discussion

Adjuvant BCG immunotherapy for prolonging RFS in intermediate/high-risk NMIBC patients has corroborated on multiple occasions its superiority when compared to endoscopic resection alone and/or adjuvant intravesical chemotherapy [83,84]. This was confirmed by individual-patient data meta-analysis [9,85] and RCT evidence comparing BCG with Epirubicin and interferon, MMC, or Epirubicin alone [50,86]. Moreover, the way BCG should be administered in terms of schedule and dose has also reached a universal uniform agreement. Recently, the phase III non-inferiority NIMBUS trial [87] has indeed failed in demonstrating reduced BCG frequency and dose to provide similar RFS in high-risk patients. This was clear with reduced schedule leading after 12 months of median follow-up to more than double the relative percent of recurrence rate, thus stopping the trial to avoid harm in the reduced frequency arm. Interestingly, the rationale from the NIMBUS trial was based on in-vivo animal trials showing robust immuno-react effects with less than the conventional number of BCG instillations. Despite not reaching the pre-established endpoints, the NIMBUS trial can be considered a comprehensive example of the logistic complexity related to the worldwide BCG supply chain yet delivering a critical message about the beneficial and life-saving effect of this medication in the arduous path of NMIBC patients. The trial was sample size re-adjusted during the course of the enrollment while it was struggling to increase the number of Institutions and patients due to the lack of BCG availability across European countries.

The concept of “*adequate BCG exposure*”, just lately introduced in the EAU Guidelines, may be interpreted as a proposed alternative for the necessity of facing the world crisis in BCG production. While the original Lamm protocol [2] consisting of a 3-year schedule is indeed the most efficient in preventing recurrence from the high-risk category, this seems nowadays challenge to apply in routine clinical practice both from a patient (due to toxicity and compliance to treatment profile) and industrial chain supply perspective.

The chance to test different BCG strains for exploring more efficient BCG properties derived from variable immunogenicity and reactogenicity in the prevention of recurrence and/or progression is, therefore, particularly appealing since no clear evidence has so far demonstrated the superiority of one strain over the other. Of note, this lack of high-quality evidence has induced the American Food and Drug Administration (FDA) to not expand the manufacturing indication towards other than non-American BCG stains, even during the hardest BCG shortage period.

The hypothesis for existing variable BCG strains efficacy is, however, concrete if we do account for the historical developments and worldwide distribution of the oral and intradermal TB vaccine. The wide variability in BCG strain genetic profiling is indeed conferred by serial laboratories passages which have led to a continuous modification of the freeze-dried seed lots nowadays cultured by different manufacturing companies across the world. Subsequent deletions of the region of difference 1 (RD1) responsible for the protein secretion system ESX-1 have provided different BCG sub-strains, which are nowadays implemented in clinical practice [4]. The interplay between these genetic differences and immuno-reactivity for clinical outcomes was tested in an RCT setting. The experience from Rentsch et al. [66] was able to highlight the longer RFS of BCG RIVM and Connaught, respectively, when compared to BCG Tice. Similar was observed in the largest European cohort study on T1 G3 NMIBCs from Witjes et al. [77], which identified BCG Connaught superiority in preventing recurrence. After adjusting for adequate confounders usually associated with recurrence in NMIBCs, the authors found a significantly longer time to first recurrence on Connaught as compared with Tice (HR, 1.48; 95%CI: 1.20–1.82). The common denominator of these experiences highlighting such different efficacy profiles was represented by the implementation of an induction-only schedule. The results were indeed the opposite when the sole sub-group of patients who had undergone maintenance was reviewed. In the study of Witjes [77], BCG Tice revealed a long-acting potential exhibiting improved efficacy outcomes in the long-term maintenance setting opposing a decrease in the immune response over time demonstrated by BCG Connaught.

In our study, we did not find any consistent differences in the relative efficacy of the BCG strains adopted both at per cumulative event rate level stratified by RFS timepoints and at the network metanalysis of the direct and indirect treatment comparisons. This was also true when comparing our outcomes to those previously published in other series [88,89,90,91]. However, at a closer analysis of our results, BCG Tokyo 172 was the sole exhibiting a relatively lower percentage of pooled BC recurrence (i.e., 0.22, 95%CI 0.16–0.27) but only in the 1 to 3 years RFS time points, yet this not translated in any BCG contrasts superiority at network assessment (Figure 3). Of note, when interpreting results from studies assessing outcomes at longer RFS endpoints (i.e., >3-y RFS), we noticed a slight but constant decrease in the recurrence rate per strain when a cumulative meta-analysis by publication year was applied. While we would be cautious in deeming conclusive conjectures, this could be interpreted as a tendency for lower efficacy of BCG over the course of follow-up, testifying an indirect loss of efficacy potentially related to lowered immunoreactivity. On the other hand, this observed phenomenon could be related to the variation in the NMIBC grading system over time (i.e., 1973 vs. 2004/2016) so that patients enrolled in the different series might not always be overlapping in terms of risk stratification. The additional explanation includes the possibility of intradermal BCG exposure prior to intravesical instillations as per the tuberculosis vaccine to which a large subset of the population in Europe had been exposed in the past four decades.

In addition, in the network meta-analysis, the rank of probable best strains for lowering recurrence was mainly identified in BCG RIVM and Tice. This is in line with what was recently documented by Del Giudice et al. [31] in a large comparative retrospective series of Connaught, RIVM, and Tice. These last two BCG strains exhibited indeed longer RFS, which was also related to a possible better tolerability profile leading to a longer course of maintenance and a total number of instillations delivered. An interesting point of debate was furthermore underlined in a subsequent multicenter experience by the same groups of authors focused on the direct comparison of the sole RIVM and Tice among high-risk NMIBCs [32,92]. What emerged at the sub-group analysis level was that Tice was able to exhibit longer RFS only in those cases where a strict adhesion to EAU guidelines recommendations was followed with regard to routinely performing secondary resection.

Our study is not devoid of limitations. First and most importantly, we would readily admit the existence of substantial single-study variability both within patients population and tumor characteristics as well as per single-strain adopted by each of the different study designs included. While indeed, NMIBC recommendations guidelines for clinical trial development [93] insist on the necessity to avoid hyper-categorization of the different confounders and cohort features which otherwise would be too far from clinical practice reproducibility, this aspect, however, significantly influenced the pooled estimates from meta-analytic calculations across a wide range of study assessed. However, we deeply attempted to reduce the single study and arm heterogeneity by constantly repeating sub-group analysis and meta-regression for the most valuable confounders, including schedule, dosage, previous BC history, and naïve vs. non-naïve intravesical patients. This led to a significant reduction in the interpretation of the displayed results to the sole BCG strains, which have been more consistently reported in the literature, while the less utilized could have been only mentioned yet not cumulatively compared. Moreover, clinical practice and Guidelines have widely changed over the range of the study period, and for this reason, imperative indications such as Re-TUR in high-risk cases could not have been captured. Additionally, our analysis was mainly focused on the outcome of BC recurrence within different BCG strains, and we were not able to cumulatively represent the same analyses on progression and cancer-specific survival due to a lack of an adequate similar number of series assessing the outcomes. However, this could have been misleading since several additional factors, such as the influence of upfront cystectomy in highest-risk patients as well as alternative strategies for BCG unresponsive patients, could have been missed. Finally, we did not report any cumulative estimates on the well-known BCG-related adverse effects and toxicity events across the arms screened nor in the single BCG strains assessment. While this was out of the scope of the current report, we would readily recognize the possible effect of such a confounder of patients censored among the studies due to side effects related to drop-out.

## 5. Conclusions

Our results do not support any clear advantage of one specific BCG strain over another. The evidence from our results is in line with previous meta-analyses, yet it updates the current state of the art by exploring time-dependent endpoints of BC recurrence rates and extrapolating the comparisons from the trials exhibiting different BCG strains as the common denominator. These findings make the decision to invest in future head-to-head RCTs on BCG strain comparison challenging while possibly diminishing the interest in the manufacturing supply chain to drive the production among strains. However, according to our pooled estimates and network results, future research could be oriented toward BCG Tokyo 172, RIVM, and Tice, which showed possible insights into efficacy profiling. This could be of particular interest, especially now that international NMIBC recommendations from various Guidelines have emphasized the importance of adequate risk-group assessment, secondary resection, as well as frequency, and dose of adjuvant BCG immunotherapy.

## Figures and Tables

**Figure 1 cancers-15-01937-f001:**
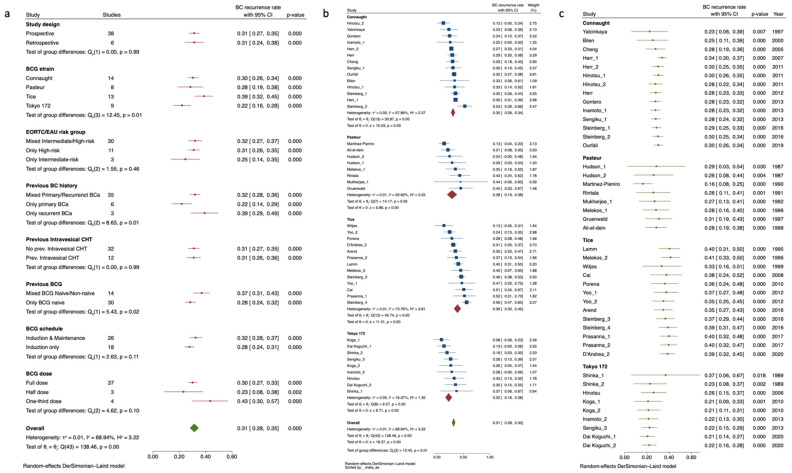
Sub-groups analysis for BC recurrence rate by studies with ≤3-y RFS endpoints exploring heterogeneity according to categorical confounders (e.g., study, BCG, and NMIBC characteristics) (**a**). Forrest plot depicting BC recurrence rate by studies with ≤3-y RFS endpoints grouped by BCG strains and sorted by increasing effect size (**b**). Forrest-plot for cumulative meta-analysis by studies with ≤3-y RFS endpoints grouped according to BCG strain and sorted by year of publication (**c**). BC: Bladder Cancer; BCG: Bacillus Calmette–Guérin; Bladder Cancer; CHT: Chemotherapy; EAU: European Association of Urology.

**Figure 2 cancers-15-01937-f002:**
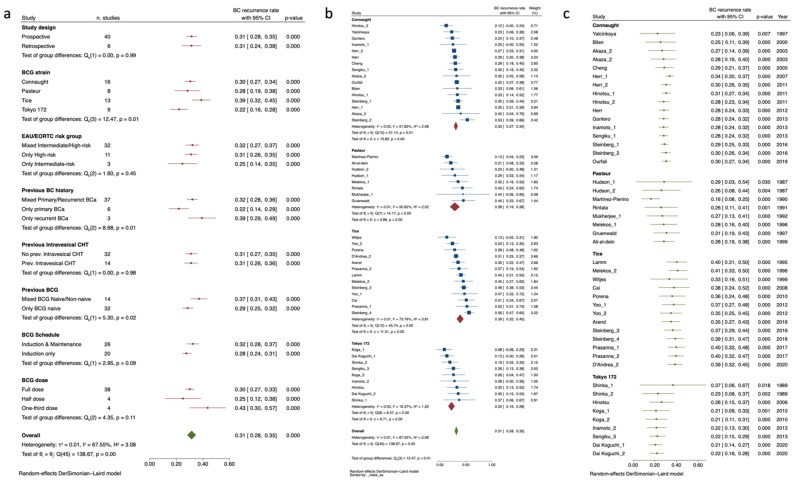
Sub-groups analysis for BC recurrence rate by studies with >3-y RFS endpoints exploring heterogeneity according to categorical confounders (e.g., study, BCG, and NMIBC characteristics) (**a**). Forrest plot depicting BC recurrence rate by studies with >3-y RFS endpoints grouped by BCG strains and sorted by increasing effect size (**b**). Forrest-plot for cumulative meta-analysis by studies with >3-y RFS endpoints grouped according to BCG strain and sorted by year of publication (**c**). BC: Bladder Cancer; BCG: Bacillus Calmette–Guérin; Bladder Cancer; CHT: Chemotherapy; EAU: European Association of Urology.

**Figure 3 cancers-15-01937-f003:**
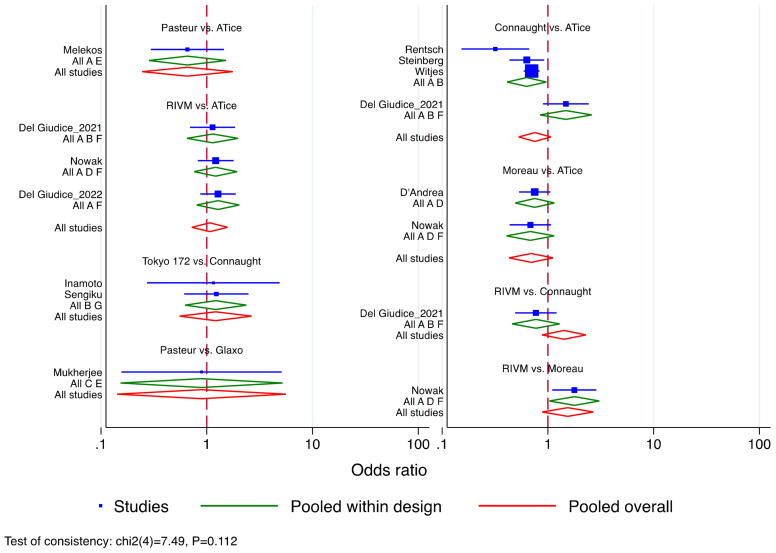
Forrest plot for bladder cancer (BC) risk of recurrence showing the individual study results grouped by treatment contrast and design.

**Table 1 cancers-15-01937-t001:** Clinical, demographic, and BCG-related features of the studies and patients enrolled in the systematic review and meta-analysis. MIBC: muscle-invasive bladder cancer; EAU: European Association of Urology; BCG: Bacillus Calmette–Guérin; yr: Year; mo: Months.

Study Author	Year	Study Design	Sample Size (n)	Median Age (yr)	Male Gender (%)	EAU Risk Group	Previous BC History	BCG Strain	BCG Schedule	BCG Dose	Previous BCG	Follow-Up (mo)
Agrawal [23]	2007	Prospective	128	64.5	71.8	Only Intermediate	Primary /Recurrent	Danish 1331	Induction & Maintenance	Full Dose vs. one-third Dose	Naïve	36
Ali-el-dein [24]	1999	Prospective	58	58.5	72.4	Intermediate/ High-risk	Primary /Recurrent	Pasteur	Induction & Maintenance	Full Dose	Naïve	30
Arend [25]	2016	Prospective	190	67.4	84.2	Intermediate/ High-risk	Primary /Recurrent	Tice	Induction & Maintenance	Full Dose	Naïve & Recurrent	24
Bilen [26]	2000	Prospective	21	53	95.2	Only High-risk	Primary /Recurrent	Connaught	Induction only	Full Dose	Naïve	18
Brosman [27]	1982	Prospective	39	63.4	73.7	Intermediate/ High-risk	Recurrent only	Tice	Induction & Maintenance	Full Dose	Recurrent	24
Cai [28]	2008	Prospective	81	69.8	86.4	Only High-risk	Recurrent only	Tice	Induction & Maintenance	Full Dose	Naïve & Recurrent	15
Cheng [29]	2005	Prospective	102	70.1	71.5	Intermediate/ High-risk	Primary /Recurrent	Connaught	Induction & Maintenance	Full Dose	Naïve	23
D’Andrea [30]	2020	Retrospective	660	63	89.7	Intermediate/ High-risk	Primary /Recurrent	Moreau/Tice	Induction & Maintenance	Full Dose	Naïve & Recurrent	41
Dai Koguchi [31]	2020	Retrospective	78	76	82.5	Intermediate/ High-risk	Primary /Recurrent	Tokyo 172	Induction & Maintenance/ Induction only	Half-dose	Naïve	35
Del Giudice [32]	2021	Retrospective	422	67	67.7	Intermediate/ High-risk	Primary /Recurrent	Connaught/RIVM/Tice	Induction & Maintenance	Full Dose	Naïve	73
Del Giudice [33]	2022	Retrospective	852	68	74.4	Only High-risk	Primary only	Tice/RIVM	Induction & Maintenance	Full Dose	Naïve	53
Dereijke [34]	2005	Prospective	84	-	94	Only High-risk	Primary /Recurrent	Connaught	Induction only	Full Dose	Naïve & Recurrent	60
Di Lorenzo [35]	2010	Prospective	40	71.4	55	Only High-risk	Recurrent only	Connaught	Induction & Maintenance	Full Dose	Recurrent	16
Di Stasi [36]	2006	Prospective	105	67	81.9	Intermediate/ High-risk	Primary /Recurrent	Connaught	Induction & Maintenance	Full Dose	Naïve	88
Farah [37]	2014	Prospective	60	61.7	83.3	Intermediate/ High-risk	Primary /Recurrent	RIVM	Induction & Maintenance	Full Dose	Naïve	48
Friedrich [38]	2007	Prospective	163	67	80.4	Intermediate/ High-risk	Primary /Recurrent	RIVM	Induction only	Full Dose	Naïve	36
Gontero [39]	2013	Prospective	120	67.5	-	Only Intermediate	Primary only	Connaught	Induction & Maintenance	One-third Dose	Naïve	12
Gruenwald [40]	1997	Prospective	40	68.5	87.5	Only High-risk	Primary /Recurrent	Pasteur	Induction only	Full Dose	Naïve	28
Hemdan [41]	2014	Prospective	126	-	79.2	Intermediate. /High-risk	Primary only	Tice	Induction & Maintenance	Full Dose	Naïve	60
Herr [42]	2012	Prospective	156	68	67.5	Only High-risk	Primary /Recurrent	Connaught	Induction only	Full Dose	Naïve	24
Herr [43]	2011	Prospective	816	64	73	Only High-risk	Primary /Recurrent	Connaught	Induction only	Full Dose	Naïve & Recurrent	24
Herr [44]	2007	Prospective	805	65	76	Only High-risk	Recurrent only	Connaught	Induction & Maintenance /Induction only	Full Dose	Naïve & Recurrent	24
Hinotsu [45]	2011	Prospective	83	-	95.2	Intermediate/ High-risk	Primary /Recurrent	Connaught	Induction & Maintenance /Induction only	Full Dose	Naïve & Recurrent	24
Hinotsu [46]	2006	Prospective	40	64.3	78	Intermediate/ High-risk	Primary /Recurrent	Tokyo 172	Induction only	Full Dose	Naïve	36
Hudson [47]	1987	Prospective	42	67	71.4	Only Intermediate	Primary only	Pasteur	Induction & Maintenance /Induction only	Full Dose	Naïve	17
Ibrahiem [48]	1988	Prospective	17	55	82.3	Intermediate/ High-risk	Recurrent only	Montreal Armand Frappier	Induction & Maintenance	Full Dose	Naïve & Recurrent	28
Inamoto [49]	2013	Prospective	38	72.5	85	Intermediate/ High-risk	Primary /Recurrent	Connaught/Tokyo 172	Induction & Maintenance	Full Dose /Half Dose	Naïve & Recurrent	12
Jarvien [50]	2009	Prospective	44	68	77.2	Intermediate/ High-risk	Recurrent only	Pasteur	Induction & Maintenance	Full Dose	Naïve	60
Kamat [51]	1994	Prospective	95	54.1	86	Intermediate/ High-risk	Primary /Recurrent	Danish 1331	Induction Only	Full Dose	Naïve	60
Koga [52]	2010	Prospective	51	74	79	Intermediate/ High-risk	Primary /Recurrent	Tokyo 172	Induction & Maintenance /Induction only	Full Dose	Naïve	29
Lamm [2]	2000	Prospective	384	67	90.1	Intermediate/ High-risk	Primary /Recurrent	Connaught	Induction & Maintenance /Induction only	Full Dose	Naïve & Recurrent	84
Lamm [53]	1995	Prospective	469	66.5	84	Intermediate/ High-risk	Primary /Recurrent	Tice	Induction & Maintenance	One-third Dose	Naïve	30
Martinez-Pieniro [54]	2002	Prospective	500	-	89.3	Intermediate/ High-risk	Primary /Recurrent	Connaught	Induction & Maintenance	Full Dose /one-third Dose	Naïve	69
Martinez-Pieniro [55]	1990	Prospective	67	65	82	Intermediate/ High-risk	Primary /Recurrent	Pasteur	Induction only	Full Dose	Naïve	36
Marttila [56]	2016	Prospective	115	69.5	74	Only Intermediate	Primary /Recurrent	Tice	Induction & Maintenance	Full Dose	Naïve & Recurrent	90
Melekos [57]	1996	Prospective	46	65.4	89.1	Intermediate/ High risk	Primary /Recurrent	Pasteur/Tice	Induction & Maintenance	Full Dose	Naïve & Recurrent	35
Mukherjee [58]	1992	Prospective	21	-	-	Intermediate/ High-risk	Primary /Recurrent	Glaxo /Pasteur	Induction only	Full Dose	Naïve & Recurrent	12
Nowak [59]	2021	Retrospective	590	71.1	85.3	Only High Risk	Primary only	Moreau/ RIVM/Tice	Induction & Maintenance	Full Dose	Naïve & Recurrent	40
Oddens [3]	2012	Prospective	1355	69	80.7	Intermediate/ High-risk	Primary /Recurrent	Tice	Induction & Maintenance	Full Dose/ one-third Dose	Naïve	84
Ojea [60]	2007	Prospective	430	-	88	Only Intermediate	Primary /Recurrent	Connaught	Induction & Maintenance	One-third Dose	Naïve & Recurrent	57
Okamura [61]	2011	Retrospective	75	68	88.8	Intermediate/ High-risk	Primary /Recurrent	Tokyo 172	Induction & Maintenance/ Induction only	Full Dose	Naïve	66
Oosterlinck [62]	2011	Prospective	48	70	81.3	Intermediate/ High-risk	Primary only	Tice	Induction & Maintenance	Full Dose	Naïve	56
Ourfali [63]	2019	Retrospective	402	-	-	Only High-risk	Primary /Recurrent	Connaught	Induction & Maintenance	Full Dose	Naïve	24
Palou [64]	2001	Prospective	126	63	95	Intermediate/ High-risk	Primary /Recurrent	Connaught	Induction & Maintenance /Induction only	Full Dose	Naïve	78
Peyromaure [81]	2003	Retrospective	57	65.4	-	Intermediate/ High-risk	Primary /Recurrent	Connaught	Induction & Maintenance	Full Dose	Naïve & Recurrent	48
Porena [65]	2010	Prospective	64	68.7	87.5	Only High-risk	Primary only	Tice	Induction & Maintenance	Full Dose	Naïve	44
Prasanna [66]	2017	Retrospective	103	77	83	Intermediate/ High-risk	Primary /Recurrent	Tice	Induction & Maintenance	Full Dose	Naïve & Recurrent	15
Rentsch [67]	2014	Prospective	131	72	83.3	Only High risk	Primary /Recurrent	Connaught/ Tice	Induction only	Full Dose	Naïve & Recurrent	51.4
Rintala [68]	1991	Prospective	51	68	76.4	Intermediate/ High-risk	Primary /Recurrent	Pasteur	Induction only	Full Dose	Naïve	28
Sekine [69]	2001	Prospective	42	72	80.9	Only High-risk	Primary /Recurrent	Tokyo 172	Induction only	Full Dose	Naïve	47
Sengiku [70]	2013	Prospective	129	70.7	76.1	Intermediate/ High-risk	Primary /Recurrent	Connaught/ Tokyo 172	Induction only	Full Dose	Naïve	28
Shinka [71]	1997	Prospective	141	-	83	Intermediate/ High-risk	Primary only	Tokyo 172	Induction only	Full Dose	Naïve	60
Shinka [72]	1989	Prospective	56	67.8	-	Only High-risk	Recurrent only	Tokyo 172	Induction only	Full Dose	Naïve	20
Sood [73]	2020	Prospective	104	58	94.1	Intermediate/ High-risk	Primary only	Sii Onco BCG	Induction & Maintenance	Full Dose	Naïve	36
Steinberg [74]	2016	Prospective	398	70	78.4	Intermediate/ High-risk	Primary /Recurrent	Connaught/ Tice	Induction & Maintenance	Full Dose /one-third Dose	Naïve & Recurrent	24
Sylvester [75]	2010	Prospective	281	66	NR	Intermediate/ High-risk	Primary /Recurrent	Tice	Induction only	Full Dose	Naïve	108
Takashi [76]	1998	Prospective	84	65.3	75	Only High-risk	Primary /Recurrent	Tokyo 172	Induction only	Full Dose	Naïve	56
Takashi [77]	1997	Retrospective	30	65.6	83.3	Only High-risk	Primary /Recurrent	Tokyo 172	Induction only	Full Dose	Naïve	60
Witjes [78]	2016	Retrospective	2099	-	82.8	Intermediate/ High risk	Primary /Recurrent	Connaught/ Tice	Induction & Maintenance	Full Dose	Naïve & Recurrent	62.4
Witjes [79]	1999	Prospective	55	-	85.7	Only High-risk	Primary only	Tice	Induction & Maintenance	Full Dose	Naïve	12.3
Yalcinkaya [80]	1997	Prospective	80	55.2	55	Intermediate/ High risk	Primary /Recurrent	Connaught	Induction only	Full Dose	Naïve	33
Yoo [82]	2012	Prospective	126	61.7	81.5	Intermediate/ High risk	Primary /Recurrent	Tice	Induction & Maintenance /Induction only	Full Dose	Naïve	24

## Data Availability

The data presented in this study are available in this article and Appendix A.

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
