# Peer review of "Efficacy of Different Bacillus of Calmette-Guérin (BCG) Strains on Recurrence Rates among Intermediate/High-Risk Non-Muscle Invasive Bladder Cancers (NMIBCs): Single-Arm Study Systematic Review, Cumulative and Network Meta-Analysis"

_cancers, 2023, doi:10.3390/cancers15071937_

Round 1

Reviewer 1 Report

This is a nicely crafted and written paper which aimed to compare through large systematic review/meta-analysis the BC recurrence rates of the currently available BCG strains in NMIBC, analyzing 84 studies which qualified for their inclusion criteria over the last 40 years.  The authors who acknowledged that they were unable to capture key endpoints like progression rates confirmed previous works supporting the absence of clinical efficacy among the different strains despite well established genetic differences.

This is not unexpected as as an anti-tuberculosis vaccine, the different BCG strains have consistently demonstrated similar protection.

Comment

The only comment relates to the constant decrease in the recurrence rate per each strain by publication year, which the authors interpreted as a tendency for lower efficacy of BCG over time possibly related to lower immunoreactivity. It might be worth noting that NMIBC grading has changed over time (WHO 1973 then 2004) so that patients might or might not always be comparable. This could be mentioned in the discussion.

2. Another explanation is that 40 years ago a large subset of the population, especially in Europe had been exposed to tuberculosis/mycobacteria. As initially BCG in the 70's was given with a intradermal exposure prior to intravesical instillations, indeed what the authors refer to "immunoreactivity" may be related to immune exposure to mycobacteria. It would be worth clarifying.

Author Response

This is a nicely crafted and written paper which aimed to compare through large systematic review/meta-analysis the BC recurrence rates of the currently available BCG strains in NMIBC, analyzing 84 studies which qualified for their inclusion criteria over the last 40 years.  The authors who acknowledged that they were unable to capture key endpoints like progression rates confirmed previous works supporting the absence of clinical efficacy among the different strains despite well-established genetic differences. This is not unexpected as an anti-tuberculosis vaccine, the different BCG strains have consistently demonstrated similar protection.

REPLY: We thank the reviewer for the kind comment and summary of our research study.  

The only comment relates to the constant decrease in the recurrence rate per each strain by publication year, which the authors interpreted as a tendency for lower efficacy of BCG over time possibly related to lower immunoreactivity. It might be worth noting that NMIBC grading has changed over time (WHO 1973 then 2004) so that patients might or might not always be comparable. This could be mentioned in the discussion.

REPLY: We would like to thank the reviewer for her/his insightful comment. This has now been amended in the discussion section addressing it as a possible limitation of our interpretation of the results.

  1. Another explanation is that 40 years ago a large subset of the population, especially in Europe had been exposed to tuberculosis/mycobacteria. As initially BCG in the 70's was given with a intradermal exposure prior to intravesical instillations, indeed what the authors refer to "immunoreactivity" may be related to immune exposure to mycobacteria. It would be worth clarifying.

REPLY: We thank the reviewer for her/his constructive criticism. Previous exposure along 70s’ to mycobacteria due to intradermal vaccine might indeed be an interesting explanation for the observed reduced efficacy profile in NMIBC recurrence outcomes. This suggestion has now been implemented in the discussion section and acknowledged as a possible confounding factor across the cohort population screened in our analysis.

Reviewer 2 Report

The authors tried to compare the Efficacy of different Bacillus of Calmette-Guérin (BCG) Strains on Recurrence Rates among Intermediate/High-risk Non-muscle Invasive Bladder Cancers (NMIBCs) and they have done a lot of work. I have several questions.

1. This study only evaluates the impact of BCG strains on recurrence but not on progression, the title is not accurate.

2. This manuscript includes studies from 1982 to 2022. When were the EAU risk groups introduced? And how did you identify risk groups in the studies that did not provide EAU risk groups?

3. Besides BCG strain tumor characteristics, there are many factors which may have an impact on the BCG treatment efficacy, such as BCG dosage, courses of treatment, re-TURBT et al. This study uses single-arm data. Were the statistical methods used in the study adequate to adjust for these confounding factors? In addition, many of the studies included in this analysis provided additional confounding factors. In addition, many studies included this analysis provided more confounding factors. For example, the study conducted by Akaza et al 1995. The patients in this study were screened by previous BCG treatment and many patients in this study did not underwent TURBT and only receive BCG treatment. This is not consistent with the author's report in the manuscript (“Sixty-four studies [2,3,23-84] were included in the systematic review with a total of n=15,451 patients who received adjuvant intravesical BCG following TURBT ± Re-TUR with EAU intermediate/high-risk NMIBCs”). Therefore, I do not believe that this study will provide convincing results.

Author Response

The authors tried to compare the Efficacy of different Bacillus of Calmette-Guérin (BCG) Strains on Recurrence Rates among Intermediate/High-risk Non-muscle Invasive Bladder Cancers (NMIBCs) and they have done a lot of work. I have several questions.

REPLY: We thank the reviewer for the kind comment and summary of our research study.  

  1. This study only evaluates the impact of BCG strains on recurrence but not on progression, the title is not accurate.

REPLY: We would like to thank the reviewer for her/his comment. We would indeed acknowledge that our only primary aim was to compare cumulative recurrence rates and its network regarding the use of different BCG strains over the course of follow-up. Progression outcomes, as addressed in the limitation section of our article, reflect to many further additional cancer-specific confounders which might not be captured with the present design of study. For this reason, the title of our article clearly specify that we will be relying on only recurrence rates.

  1. This manuscript includes studies from 1982 to 2022. When were the EAU risk groups introduced? And how did you identify risk groups in the studies that did not provide EAU risk groups?

REPLY: We would like to thank the reviewer for her/his insightful comment. Risk groups within the EAU Guidelines have indeed been introduced in the 2008 update available at: Babjuk, Marko, et al. "EAU guidelines on non-muscle-invasive urothelial carcinoma of the bladder." European urology 54.2 (2008): 303-314. However, as nicely identified by the reviewer, the risk group stratification relied on the EORTC risk tables from the article of Sylvester RJ, et al. “Predicting Recurrence and Progression in Individual Patients with Stage Ta T1 Bladder Cancer Using EORTC Risk Tables: A Combined Analysis of 2596 Patients from Seven EORTC Trials”. European Urology 49: 466 - 477, 2006. We would therefore readily underline the possible confounding terminology adopted in our methods section when we addressed the EAU risk groups stratification criteria we adopted. The text has now been amended to reflect the multiple adoption of the classification system deriving from the EORTC risk tables for early studies and the EAU risk group in the most recent series. Additionally, as we would readily recognize the absence for such standardized criteria in case of earlier experiences, we relied on the punctual examination of the NMIBC cohort assessed in these series and considered eligible only those for which a currently validated “intermediate” or “high-risk” group would have been assigned according to current standard EAU definitions. This approach is in line to previous meta-analysis published on the same topic which had to deal to the aforementioned issue. However, in order to address this possible confounding factor, we re-run a set of cumulative recurrence rate meta-analysis with a temporal time established at the first series after 2008 and we did not find any discrepancies both in pooled estimates and in the heterogeneity assessment thus validating our inclusion methodological criterion.

The text has now been amended to reflect such methodological process of enrollment.

  1. Besides BCG strain tumor characteristics, there are many factors which may have an impact on the BCG treatment efficacy, such as BCG dosage, courses of treatment, re-TURBT et al. This study uses single-arm data. Were the statistical methods used in the study adequate to adjust for these confounding factors?

REPLY: We would like to thank the reviewer for her/his insightful comment. This is indeed a critical point of our analysis. In each population-based cohort study, and in general in many NMIBC studies, the assessment of literature-based risk factors is foremost important in the assessment of survival analysis and in our case for BCa recurrence assessment. A major point of strength of the present analysis is indeed the multiple evaluation of sub-groups analysis by pre-determined risk factors such as maintenance schedule, BCG dose, risk groups etc. Additionally, all possible continuous confounders such as the relative percentages of gender, smokers etc. were systematically annotated and included in our meta-regression analyses available in the supplementary materials. This search assessment has bene done on all the finally included articles and analyzed in all those studies which made available the pre-specified variables for the cohort of interest. Additionally, this heterogeneity evaluation was  possible in more than the two-third of the studies included thus justifying the validity of the proposed methodology and the derived results. 

In addition, many studies included this analysis provided more confounding factors. For example, the study conducted by Akaza et al 1995. The patients in this study were screened by previous BCG treatment and many patients in this study did not underwent TURBT and only receive BCG treatment. This is not consistent with the author's report in the manuscript (“Sixty-four studies [2,3,23-84] were included in the systematic review with a total of n=15,451 patients who received adjuvant intravesical BCG following TURBT ± Re-TUR with EAU intermediate/high-risk NMIBCs”). Therefore, I do not believe that this study will provide convincing results.

REPLY: We would like to thank the reviewer for such detailed attention in the series have included in our study. We would indeed readily acknowledge that the study from Akaza et al (1995) relied on an intention to treat analysis not matching with our definition for inclusion criterion. This is clearly a limitation of systematic review and meta-analysis which do include a wide range of study time especially in a so multifaced condition such as NMIBC. However, our primary attempt was to comprehensively assess the recurrence rate trends over considerable time of BCG exposure so as to possibly comment on exposure efficacy variation over few decades of BCG treatment in NMIBCs. This is in line to NMIBC Guidelines which recommend avoiding hyper-categorization of different cohorts in order to not be too far from daily clinical practice. Nonetheless, the results displayed in the Akaza analysis did not alter the impact on cumulative estimates and/or the heterogeneity assessment. For this reason, we have now clearly stated this particular aspect in the descriptive study paragraph of the results section (i.e., Results 3.3) so as to underline the right and insightful reviewer criticism while we also amended the methods section.

Reviewer 3 Report

The systematic review presented in the paper “Efficacy of different Bacillus of Calmette-Guérin (BCG) Strains on Recurrence Rates among Intermediate/High-risk Non-muscle Invasive Bladder Cancers (NMIBCs): Single-arm Study Systematic Review, Cumulative and Network Meta-analysis” is well written and, from my point of view, would be of interest for the readers of Cancers. In spite of this, and before its publication, I would recommend the authors to perform the following changes:

Introduction: please add a short description of the paper structure.

In lines 135-136 it is said: “We performed a systematic review of the literature in PubMed, Scopus, Web of Sci-135 ence, Embase, and Cochrane from 1982 to November 2022, without language restriction”. I can understand that search was perform without language restriction, but I am sure that results were not in all the languages of the world. Please, specify in which languages were the articles obtained with such search. As a secondary issua, are you sure you were able to understand?

In lines 150-151 it is said: “comprehensive list of primary 150 and secondary fields of search criteria has been presented in Suppl. Tab. 1”. I cannot see such table on the manuscript but also, from my point of view, such table should be in the main body of the article.

In section 3.1 Search Results a flowchart would be really of interest in order to understand the process in this flowchart the number of articles at the beginning and at the end of such process should be reported.

Please check the whole text and you will see that some references are bolded. Remove it.

References: please check manuscript and do not use double-spaced lines.

Figure 1: the quality of the figure makes impossible to read it. The same happen with figure 2.

Author Response

The systematic review presented in the paper “Efficacy of different Bacillus of Calmette-Guérin (BCG) Strains on Recurrence Rates among Intermediate/High-risk Non-muscle Invasive Bladder Cancers (NMIBCs): Single-arm Study Systematic Review, Cumulative and Network Meta-analysis” is well written and, from my point of view, would be of interest for the readers of Cancers. In spite of this, and before its publication, I would recommend the authors to perform the following changes:

REPLY: We thank the reviewer for the kind comment and summary of our research study.  

Introduction: please add a short description of the paper structure.

REPLY: The introduction has been amended accordingly.  

In lines 135-136 it is said: “We performed a systematic review of the literature in PubMed, Scopus, Web of Sci-135 ence, Embase, and Cochrane from 1982 to November 2022, without language restriction”. I can understand that search was perform without language restriction, but I am sure that results were not in all the languages of the world. Please, specify in which languages were the articles obtained with such search. As a secondary issua, are you sure you were able to understand?

REPLY: We would like to thank the reviewer for such careful comment regarding on our search strategy. There were only a minor proportion of articles in native language. However, out of these, all presented the associated English version translated. Therefore, all the studies screened and lastly included in the analysis were easily accessible in English. We have now amended the text to avoid misleading interpretation of our search methodology and to improve the clarity of our methods section.

In lines 150-151 it is said: “comprehensive list of primary 150 and secondary fields of search criteria has been presented in Suppl. Tab. 1”. I cannot see such table on the manuscript but also, from my point of view, such table should be in the main body of the article.

REPLY: We would like to thank the reviewer for her/his insightful comment. The table has been labeled as Supplementary Table 1 and it summarizes the primary and secondary fields of search for our work. We have now made sure that the table is accessible for your revision. However, while we would readily recognized the importance of this section in systematic review, due to space limitation issues and editorial necessities we would prefer keep this table supplementary but however easily accessible to the readers.

In section 3.1 Search Results a flowchart would be really of interest in order to understand the process in this flowchart the number of articles at the beginning and at the end of such process should be reported.

REPLY: Similarly to previous comment, we would surely recognize the importance of such search criteria in the development of a PRISMA-based systematic review. However, given the huge volume of figures and tables deriving from our cumulative and sensitivity analyses, we have preferred improving the visibility of our pooled estimates, sub-groups and meta-regression analyses by prioritizing these ones in the main document. PRISMA flow chart is however easily accessible for the readers in the supplementary materials as Supplementary Figure 1.

Please check the whole text and you will see that some references are bolded. Remove it.

References: please check manuscript and do not use double-spaced lines.

REPLY: We would like to thank the reviewer for her/his insightful comment. The text has been amended accordingly.

Figure 1: the quality of the figure makes impossible to read it. The same happen with figure 2.

REPLY: We apologize for such inconvenience. We have now double-checked with the editorial office the quality of our figures included that should now be as sharp as possible.

Reviewer 4 Report

In this study, the authors tried to provide the most recent cumulative literature research on the topic of Bacillus of Calmette-Guérin (BCG) comparison through a cumulative meta-analysis of event rate for bladder cancer (BC) recurrence assessed at different recurrence-free survival (RFS) endpoints and subsequently a meta-analysis of comparison of different treatments of most highly adopted BCG strains. The authors concluded that their results did not support any clear advantage of one specific BCG strain over another.

Comments

This is an interesting study. The reviewer has only some minor concerns as follows:

1. Some recent studies for BCG on BC are available for reference by the author: (1) Jiang and Redelman-Sidi, BCG in Bladder Cancer Immunotherapy. Cancers (Basel). 2022 Jun 23;14(13):3073. doi: 10.3390/cancers14133073. (2) Moon, et al. Effects of intravesical BCG maintenance therapy duration on recurrence rate in high-risk non-muscle invasive bladder cancer (NMIBC): Systematic review and network meta-analysis according to EAU COVID-19 recommendations. PLoS ONE 2022;17(9):e0273733. https://doi.org/10.1371/journal.pone.0273733.

2. Some rows in Table 1 are too crowded, please check and display separately.

3. In Figures 1 and 2, the three parts (a-c) within figures are too small, difficult to read, and the fonts are inconsistent, so it is recommended to present them separately.

4. In lines 589-615, the authors did not provide the information.

Author Response

In this study, the authors tried to provide the most recent cumulative literature research on the topic of Bacillus of Calmette-Guérin (BCG) comparison through a cumulative meta-analysis of event rate for bladder cancer (BC) recurrence assessed at different recurrence-free survival (RFS) endpoints and subsequently a meta-analysis of comparison of different treatments of most highly adopted BCG strains. The authors concluded that their results did not support any clear advantage of one specific BCG strain over another.

REPLY: We thank the reviewer for the kind comment and summary of our research study.  

This is an interesting study. The reviewer has only some minor concerns as follows:

  1. Some recent studies for BCG on BC are available for reference by the author: (1) Jiang and Redelman-Sidi, BCG in Bladder Cancer Immunotherapy. Cancers (Basel). 2022 Jun 23;14(13):3073. doi: 10.3390/cancers14133073. (2) Moon, et al. Effects of intravesical BCG maintenance therapy duration on recurrence rate in high-risk non-muscle invasive bladder cancer (NMIBC): Systematic review and network meta-analysis according to EAU COVID-19 recommendations. PLoS ONE 2022;17(9):e0273733. https://doi.org/10.1371/journal.pone.0273733.

REPLY: We thank the reviewer for her/his insightful comment. The text has been amended to implement these suggested references which are in line with the scope of the present work. We thank once more for helping improving the quality of our document.

  1. Some rows in Table 1 are too crowded, please check and display separately.

REPLY: We thank the reviewer for such careful comment. However, this is the format of Cancer for tables, and we will have to rely on their formatting.

  1. In Figures 1 and 2, the three parts (a-c) within figures are too small, difficult to read, and the fonts are inconsistent, so it is recommended to present them separately.

REPLY: While we would truly thank the reviewer for her/his constructive suggestions, this issue is mainly related to editorial requirements for figures. This is a common problem of huge forest plots in meta-analysis. While the fonts have now been unified, we will be reticent in dissembling the forest plots figures since this is in line with the description within its relative paragraph of the results section and can be of help for the readers in order to cluster the information.

  1. In lines 589-615, the authors did not provide the information.

REPLY: Thank you once more for helping us perfectioning our work. This section has now been filled accordingly.

Round 2

Reviewer 2 Report

Thanks for the explanation and revision. This study took a lot of effort from the authors.

It is well known that a systematic review uses repeatable methods to find, select, and synthesize all available evidence. It answers a clearly formulated research question and explicitly states the methods used to arrive at the answer. Whether a study should be included is depend on whether it can help answer the question rather than try to include as many studies as possible. The goal of the authors and this study is to compare current evidence within all the available retrospective/prospective and/or single-/multicenter cohort studies applying different BCG strains in the adjuvant setting of patients treated by trans-urethral resection of bladder tumor (TURBT) followed or not followed by secondary resection (Re-TUR).” Clearly the study from Akaza et al (1995) does not fit their goal and only increases confounding factors. And the treatment method by Akaza was very different form current clinical practice and guidelines. This study and studies that using BCG as single treatment should not be included. And the selection criteria of this study are not well defined on this aspect. Until this question is adequately addressed, they cannot arrive at reliable and repeatable results and answers.

Author Response

Please see the attached reply. 

Round 3

Reviewer 2 Report

Thanks for the revision.

I would like to recommend the authors to recheck all the studies that they included. Clearly, the study by Akaza et al (1995) which has been removed is not the only study that has the problem. For example, many patients in the study Akaza et al (2003) also received BCG as the single treatment.

Author Response

Please see attached our reply. 
